# Sustainable Rigid Polyurethane Foam from Wasted Palm Oil and Water Hyacinth Fiber Composite—A Green Sound-Absorbing Material

**DOI:** 10.3390/polym14010201

**Published:** 2022-01-04

**Authors:** Nathapong Sukhawipat, Laksana Saengdee, Pamela Pasetto, Jatupol Junthip, Ekkachai Martwong

**Affiliations:** 1Division of Polymer Engineering Technology, Department of Mechanical Engineering Technology, College of Industrial Technology, King Mongkut’s University of Technology North Bangkok, Bangkok 10800, Thailand; 2Institut des Molecules et Materiaux du Mans, UMR CNRS 6283, Le Mans Universite, CEDEX 9, 72085 Le Mans, France; laksana.sd@gmail.com (L.S.); Pamela.Pasetto@univ-lemans.fr (P.P.); 3Faculty of Science and Technology, Nakhon Ratchasima Rajabhat University, Nakhon Ratchasima 30000, Thailand; jatupol.j@nrru.ac.th; 4Division of Science, Faculty of Science and Technology, Rajamangala University of Technology Suvarnabhumi, Phra Nakhon Si Ayutthaya 13000, Thailand

**Keywords:** used palm oil, water hyacinth fiber, sound-absorbing material, polyurethane foam

## Abstract

A novel rigid sound-absorbing material made from used palm oil-based polyurethane foam (PUF) and water hyacinth fiber (WHF) composite was developed in this research. The NCO index was set at 100, while the WHF content was set at 1%wt with mesh sizes ranging from 80 to 20. The mechanical properties, the morphology, the flammability, and the sound absorption coefficient (SAC) of the PUF composite were all investigated. When the WHF size was reduced from 80 to 20, the compression strength of the PUF increased from 0.33 to 0.47 N/mm^2^. Furthermore, the use of small fiber size resulted in a smaller pore size of the PUF composite and improved the sound absorption and flammability. A feasible sound-absorbing material was a PUF composite with a WHF mesh size of 80 and an SAC value of 0.92. As a result, PUF derived from both water hyacinth and used palm oil could be a promising green alternative material for sound-absorbing applications.

## 1. Introduction

Polyurethane foam (PUF) is widely used as a sound-absorbing material due to its light weight, ease of manufacture, and tunable properties [1,2]. It has the ability to absorb undesired sound, a serious issue with building structures, affecting human dwellings and comfort [3]. PUF is typically synthesized through the chemical reaction of polyol and isocyanate to form urethane linkage in the presence of a blowing agent. Petroleum-based polyol has been commonly used in the PUF industry. However, the limitations, air pollution, and environmental issues are taken into account. To address these issues, renewable materials including lignin [4], natural rubber [5], and starch [6] could be used as sustainable, and green polyol sources. However, the preparation process is complicated. As a consequence, many researchers are attempting to discover new materials that are easier to produce. Vegetable oils such as soybean oil [7], castor oil [8], rapeseed oil [9], soybean oil [10] and palm oil [11] are promising candidates for PUF synthesis.

Palm oil (PO), which represents the largest global production of vegetable oil, is a low-cost feedstock for PUF preparation and was tried as a replacement for the petroleum-based precursors. Fresh palm oil has been used as a polyol source because of its high activity and easy operation. Chuayjuljit, Sangpakdee and Saravari [12] developed PUF by utilizing PO as the polyol. The stiff PUF was produced, and a closed cell was discovered. Tanaka et al. [13] developed PUF by combining PO-based polyol, polyethylene glycol (PEG) or diethylene glycol (DEG), and an isocyanate. Because PO-based polyol is a soft segment, it has been claimed that PUF containing a high proportion of PO-based polyol is more flexible. Saifuddin et al. [14] used microwave to make PUF from PO. The finished PUF was hard and rigid. PUF has fairly poor characteristics, which can be enhanced by adding fruit branches and cellulose fiber. ThePO-based polyol can even be used as a soft segment of PUF.Palm oil is normally used for cooking. To avoid competition between the uses of palm oil as a food and as a polyol, used palm oil (UPO) is proposed. The UPO benefits include not only reduced environmental waste, but also value-added waste [14,15,16]. Riyapan et al. [17] prepared PUF from UPO via simultaneous epoxidation and a ring-opening reaction. The PUF sample had low sound absorption coefficients due to its closed-cell and large-cell structure. As a result, the PUF with open-cell and small-cell structures is necessary for effective sound absorption material.

To improve the properties of PUF, organic and inorganic materials have been introduced [10,18]. Although they can improve the properties, their use can lead to pollution and adverse health effects. Therefore, many researchers have attempted to use natural fiber-based cellulose as an additive. According to Berardi and Iannace [19] natural materials such as kenaf, wood, hemp, coconut, straw, and wool have a significant potential to be used in sound absorption applications. Ekici et al. [20] explored the PUF composite with tea leaf addition. The SAC was reported to be enhanced to 0.39 by the addition of 8%wt tea leaf fiber. Jian et al. [21] combined PUF with corn straw powder. The mechanical properties were improved, but close-cell structures were still observed. Chen and Jiang [22] prepared a PUF by adding bamboo leaf particles. It has the potential to improve the characteristics of PUF. Tao, Li and Cai [23] investigated how rice straw fiber and wheat straw fiber affected the sound absorption properties of rigid polyurethane foams. It was discovered that a 5% additive provided effective sound absorption and had a higher open cell. As a consequence, it was discovered that open cell PUF with a low density had significant sound absorption. Thus, incorporating low-density natural-based cellulose into PUF has a promising future. The finding of new low-density cellulose-based material for adding to PUF is the focus.

Water hyacinth fiber (WHF) is stated to be a cellulosic material with low density, high absorption, and a great potential for composite use [24]. WHF is derived from water hyacinth (*Eichhornia crassipes*), a free-floating aquatic plant found worldwide. It has become an environmental issue due to the rapid depletion of minerals and oxygen from water [25]. However, the porous interior structure of the fiber results in a low-density and it has a good prospect for enhancing the characteristics of composite materials. Saratale et al. [26] reported that water hyacinth fiber (WHF) has low-density and high mechanical properties. Abral et al. [27] validated the mechanical and physical characteristics of WHF once again. It can be used as a polymer addition to increase mechanical and absorption properties. According to the findings, WHF has the potential to be employed as an addition for numerous polymers, including polyester [27] and poly(lactic acid) [28]. Furthermore, the absorption characteristics of WHF are unique due to the porous interior. As a result, it has the absorption applications, in particular the application for heavy metal removal [25]. Sound absorption is another one-of-a-kind use. Setyowati et al. [29] prepared the sound absorption from the water hyacinth and coconut husk based fiber reinforced polymer (FRP) panel. The sound absorption increased accordingly to an SAC of above 0.7. Therefore, many researchers have attempted to use this cellulose as an additive for composite material. For this reason, the focus of this work is on the use of WHF as a PUF additive.

Therefore, this research developed a green rigid sound-absorbing material by combining UPO-based PUF and WHF. The effects of the WHF size on the mechanical properties, morphology and the sound absorption coefficient (SAC) were investigated.

## 2. Materials and Methods

### 2.1. Materials

Used palm oil for preparing recycled palm oil (RPO) was purchased from a Nonthaburi local market, Nonthaburi province (acid value of 1.41 mg KOH/g and iodine value of 40.1 mg I_2_/g). The used palm oil was first filtered before modification. Hydrogen peroxide (H_2_O_2_, 35%) and sodium hydrogen carbonate (NaHCO_3_, >90%) were purchased from Ajax Finechem (Sydney, Australia). Formic acid (HCOOH, 98%) was purchased from Fisher Chemical (Shanghai, China). Ethyl acetate (CH_3_COOC_2_H_5_, >99%) was purchased from RCI Lab-Scan Limited (Bangkok, Thailand). Polymeric diphenylmethane diisocyanate (P-MDI, 31.5% NCO content, functionality = 2.7) was purchased from BASF (Ludwigshafen, Germany). T-12 (dibutyltin dilaurate, 95%) was purchased from Fluka Chemie AG CH-9471 Company (Darmstadt, Germany). Dabco 33LV (33% triethylene diamine in propylene glycol) was purchased from Evonilk Goldschmidt GmbH (Essen, Germany). Silicone surfactant (TEGOSTAB^®^ B8110) was obtained from Gold-Schmidt (Berlin, Germany). Dried water hyacinth was purchased from Suphanburi province, Thailand. Commercial polyurethane foam, Polyurethane foam/glass fiber, and polyester foam were purchased from a convenience store, Nonthaburi province, Thailand.

### 2.2. Methods

#### 2.2.1. Preparation of Green Polyol from UPO

The green polyol was made from used palm oil (UPO). Both epoxidation and ring-opening reactions were carried out in a single step. Before the reaction, 250 g (0.3 mol) of used palm oil was filtered and dried at 70 °C for 8 h. Dropwise, 57.71 mL (1.5 mol) formic acid was introduced to a 2-L reactor containing the UPO, followed by 50.25 mL hydrogen peroxide (0.75 mol). The mixture was stirred for 4 h at 70 °C at a controlled speed of 200 rpm. The UPO-based polyol was washed with ethyl acetate, saturated NaHCO_3_ solution, and NaCl solution, respectively. To obtain pure green polyol or RPO, the sample was evaporated in a rotary evaporator at 40 °C. The iodine value, acid value, and hydroxyl value of the resulting polyol were all determined.

#### 2.2.2. Preparation of WHF Fiber

WHF with mesh sizes of 80, 40, and 20 were provided by starting with a 1 cm length of dried water hyacinth ground with a double blade blender (Thai grinder, Thailand). After grinding, the fine fiber was sieved with 80 mesh, followed 40 mesh, and finally 20 mesh. Before use, all sieved fiber samples were vacuum-dried at 80 °C for 12 h.

#### 2.2.3. Preparation of UPO-Based PUF/WHF Composite

PUF/WHF composites with an NCO index of 100 were prepared in a single step. Table 1 shows the ingredients for all filled foams. A magnetic stirrer set at 250 rpm was used to thoroughly mix the mixtures of RPO, Dabco 33LV, distilled water, and surfactant. WHF was then added to the mixture. The WHF amount was set at 1 mol, and the size of the WHF was varied from 80, 40, and 20 mesh to obtain PUF-WHF-80, PUF-WHF-40, and PUF-WHF-20, respectively. Then, PMDI was added and stirred until the liquid was white. Finally, the completed mixture was transferred to an open mold with a volume of 10 × 10 × 5.5 cm^3^ in generating free-rise foam. To finish the polymerization reaction, the PUF composites were completely cured in an oven at 50 °C for 48 h. In this study, neat PUF was made without the addition of WHF to compare with PUF composites.

Figure 1 presents a short overview of the PUF composite preparation process.

### 2.3. Characterization

The iodine values of the RPO were determined by titration to assess the double bonds in the structures, according to ISO3961-2009. The acid values were titrated to analyze the free fatty acids in the oils, according to ISO660-2009. The OH values were titrated to determine the number of OH units in the structures, according to ISO14900-2001.

The chemical structures of UPO and RPO were analyzed by ^1^H-NMR with a Bruker 400 Fourier transform spectrometer (Bruker, Berlin, Germany) at 400.13 and 100.62 MHz. All the samples were dissolved in CDCl_3_, using tetramethylsilane (TMS) as the internal standard. Fourier Transform Infrared (FTIR) spectra were recorded with a Nicolet Avatar 370 DTGS FTIR spectrometer (LabX, Midland, ON, Canada) using the range 4000–400 cm^−1^.

The molecular weights of the UPO and RPO samples were analyzed by size exclusion chromatography (SEC) using a Shodex GPC KF-806M column (Shodex, Tokyo, Japan).

The density of the PUF was determined using the ISO4590-2002 test. A Vernier caliper was used to measure the exact dimensions. The density of the specimens was calculated using the equation density = mass/volume.

The compressive stress was measured using Testometric (M500-25AT) on an Instron universal testing machine in accordance with the ISO844-2007 standard (Testometric, Rochdale, UK). All the PUF samples were cut to 50 × 50 × 30 mm^3^ size. The crosshead moved at a rate of 2.5 mm/min. The compressive strength was determined using the conventional 10% deformation method. Three duplicates of each sample type were examined, and the average findings were reported in kilopascals (kPa).

The morphology of the PUF composites was examined using a scanning electron microscope (SEM) equipped with a JSM-6510LV (JEOL, Japan) under high vacuum and high voltage settings at 20.00 kV. Before imaging, all of the samples were gold-coated. In addition, to check the structure of the WHF, an optical microscope with a magnification of 5× and Xenon (DN-117M) (Nanjing Jiangnan Novel Optics, Nanjing, China) was utilized.

The moisture absorption of the PUF/WHF composite was studied by placing the sample into the desiccator with a controlled of 60% relative humidity. It was determined in five samples for each formula. The PUF/WHF composite was weighed every day. The moisture absorption rate was calculated by the following Equation (1).
A (%) = W_f_ − W_i_/W_i_ × 100(1)
when: A (%) is absorption percentage, W_i_ is the initial weight of the PUF/WHF, and W_f_ is the weight after obtaining the moisture of the PUF/WHF.

The flammability test was performed according to ASTMD4986. This method is employed for testing the extent and time of the burning of cellular polymeric materials. The foam specimens were burned in a horizontal position with a methane burner. The standard test specimens were 50 × 150 × 13 mm^3^, with the heights 25, 60, and 125 mm marked on them with lines. The time was recorded when the flame reached the 25, 60 and 125 mm marks, and when the specimen extinguishes.

The acoustic characteristics of PUF were investigated in terms of the Sound Absorption Coefficient (SAC) according to ASTM E1050 90, utilizing Kundt’s tube, which included an impedance tube, two microphones, and a frequency analyzer (Impedance Measurement Tube Type 4206) (Brüel&Kjr, Nærum, Denmark). The impedance tube equipment contained the B&K Type 1405 Noise generator, the B&K Type 2406 Impedance tube filter and speaker, the B&K Type 2706 power amplifier, the B&K Type 4135 0.25” condenser microphone, a 0.25” microphone calibrator, the B&K Type 2406 small sample tube, the B&K Type 4206 large sample tube, and the 01 dB SYMPHONIE data acquisition hardware. The foam samples were cut to 25 mm diameter and 15 mm thickness and tested with a working frequency range of 500 to 6000 Hz. In this study, commercial polyurethane foam, polyurethane foam/glass fiber, and polyester foam were compared. The SAC was defined as the ratio of the acoustic energy absorbed by the PUF composites (I_incident_ − I_reflection_) to the incident acoustic energy (I_incident_) on the surface, as given in Equation (2).
Sound absorption coefficient (SAC), α = (I_incident_ − I_reflection_)/I_incident_(2)
where I_incident_ is the incident acoustic energy andI_reflection_ is the reflected acoustic energy.

## 3. Results

### 3.1. Characteristic of WHF

Figure 2 shows the physical appearance and optical microscope images of the WHF with different mesh sizes—80, 40, and 20 mesh. The WHFs are a fine, yellowish fiber. The OM images revealed that the WHF fiber of mesh 80, 40, and 20 had a length of 180 µm, 400 µm, and 840 µm, respectively, with a diameter of 20–25 µm. It is relevant to the standard size of the material after sieving in each mesh size. The OM images revealed that the WHF has a porous interior, and this is the point at which WHF should be introduced into the PUF. According to the idea, a material with a high porous structure and a low density can increase the sound absorption qualities [11]. As a result, WHF is one of the best materials for improving sound properties in this study.

### 3.2. UPO and UPO-Based Polyol

#### 3.2.1. Properties of UPO and UPO-Based Polyol

Table 2 lists the specific properties of the UPO and the RPO precursors of PUF/WHF composites. After conversion of UPO to RPO, the iodine number, which indicates the quantity of double bonds on the oil structure, reduced from 40.1 to 0.51. The OH values, which indicate the presence of a hydroxyl group on the structure of UPO and RPO, increased from 0 to 192.19 mg. KOH/g. After the process, the acid number related to the free acid in the UPO and the residual acid increased slightly from 1.41 to 1.76 mg. KOH/g. These findings revealed that the UPO was effectively changed by oxidation and hydroxylation to form UPO-based polyol, which corresponded to Riyapan et al. [17].

Furthermore, the molecular weight of the UPO and UPO-based polyol was measured using the SEC method. The SEC result is presented in Figure 3. It was found that the molecular weight of polyol-based UPO is higher than that of UPO. Because of the breakage of a double bond to produce OH functional groups on the polyol structure, the *M_n_* value rose from 2841 to 3073 g/mol. The polydispersity index (PDI) dropped from 1.08 to 1.02. Furthermore, the FTIR and ^1^H-NMR methods were used to validate the chemical structures and functional groups of the UPO and UPO-based polyol.

#### 3.2.2. Chemical Structure Confirmation

The chemical structure of UPO and polyol-based UPO was confirmed by FTIR and ^1^H-NMR. Figure 4 presents their spectra results. Figure 4a illustrates the ^1^H-NMR spectra results of UPO and UPO-based polyol. The vital peaks at 4.2, 5.1, and 5.3 ppm were assigned to 1, (–C**H**_2_O(C=O), 2 (–C**H**O(C=O), and 7, (–C**H**=C**H**–), respectively. This implied the typical structure of triglyceride. Following the alteration, additional peaks at 3.0 ppm corresponding to methine protons (11, (–CH_2_C**H**(OH)) were discovered. Furthermore, a peak at 5.3 ppm disappeared due to the absence of the double bond in the triglyceride structures. These findings verified the effective synthesis of the UPO-based polyol.

Figure 4b presents the FTIR spectra of UPO and UPO-based polyol. The vibration bands of the UPO sample were found at 1150, 1600, 2800–3000, and 3100 cm^−1^, which were attributed to the C–O–C, C=O, C–H, and =CH double bonds of triglyceride functional groups, respectively. After the modification, a new broad peak at 3300–3500 cm^−1^ was observed, indicating the existence of the OH group.

As a result of these chemical structural results, the OH group on the UPO-based polyol may be employed as a green polyol to manufacture the PUF and PUF/WHF composites.

### 3.3. PUF/WHF Composite

#### 3.3.1. Properties of PUF/WHF Composite

Table 3 lists the properties of the PUF/WHF composite. The PUF was compared to the PUF/WHF composites in the blank test. The PUF had a cream time of 12, a rise time of 23, and a track free time of 1282 s. The cream time and the rise time for the PUF/WHF sample rose as the WHF size increased. The cream time and the rise time of the PUF-WHF-80 with the smallest size of WHF were 13 s and 27 s, respectively, whereas the cream time and the rise time of the PUF-WHF-20 were 14 and 35 s. The track-free time of the PUF/WHF composite, on the other hand, reduced as the WHF size increased. The PUF-WHF-80 had a track-free time of 1255 s, while the PUF-WHF-20 had a time of 1154 s. These findings suggested that the size of the WHF might inhibit the production of PUF during the creation of urethane linkages.

The heights of the PUF, PUF-WHF-80, PUF-WHF-40, and PUF-WHF-20 were 8.4, 8.0, 7.5, and 6.2 cm, respectively. The results showed that the WHF with a large size might reduce the track-free time due to the restriction of the foam growth. Furthermore, the mechanical characteristics of the WHF were superior to those of the PUF due to the greater limit of the PUF expansion. As a result, WHFs with a large size lowered the expansion of the PUF composite and height values.

Another property was the density values. The presence of the WHF reduced the density. Long fiber has a higher density than short fiber, whereas WHF has a lower density than neat PUF. As a result, the density of PUF-WHF-20 is higher than PUF-WHF-80 but lower than that of neat PUF.

The hardness (shore OO) of the PUF composites was also measured. The hardness values of the PUF, PUF-WHF-80, PUF-WHF-40, and PUF-WHF-20 were 29, 45, 37, and 33, respectively. The results revealed that when fine fiber was introduced, the hardness of the PUF composites increased. It is due to the reinforcing effects of the additives. When a smaller fiber with a higher surface area was introduced, the increased hardness was exhibited.

The mechanical characteristics of the PUF/WHF composite were reported in terms of compression strength, as illustrated in Figure 3. The compression strengths of the neat PUF, PU-WHF-80, PU-WHF-40, and PU-WHF-20 were 0.027, 0.033, 0.042, and 0.047 kPa, respectively. This finding indicates that a large size of WHF could increase the compression strength of the PUF/WHF composite, which was relevant with the foam expansion. It is possible that chain entanglement of long fiber might form between the long fibers.

#### 3.3.2. Morphology of PUF/WHF Composite

Figure 5 illustrates the morphological results of the PUF composites. The neat PUF had a closed-cell structure with a large cell size ranging from 0.20 to 1.30 mm. The addition of the WHF resulted in the formation of open-cell foam. During foam expansion, the interface between the WHF and the PUF matrix is weak, and the WHF damages the closed cells of the PUF. As a result, adding WHF to PUF results in more open cells than that of neat PUF. This result is related to the work of Tao et al. [23] and Członka et al. [10]. However, because the strength of the long fiber was greater than the PUF, the biggest additive fiber, PUF-WHF-20, could not expand as much as it could. The short fibers, WHF-40 and WHF-80, on the other hand, may be well-dispersed in the PUF matrix and generate more open cells for the PUF. The distribution of cell foam in smaller sizes was observed as the short fiber was filled. As a result, the addition of WHF-80 resulted in more open cells and a reduction in the cell size of the PUF composite. Furthermore, it was discovered that when a small size of WHF was applied, the regularity of the cell foam and cell size distribution improved. One of the purposes of this work is to increase the sound absorption capabilities of PUF by using an open cell and a small pore size [30]. As a result, the aim of the PUF composites with a small size and more open cells is achieved.

#### 3.3.3. Moisture Absorption of PUF/WHF Composite

Moisture absorption was also studied in order to evaluate the behavior of the materials over the use term. In comparison to the real condition, the relative humidity was controlled at 60%. Figure 6 demonstrates the moisture absorption of the PUF compared to the PUF composite. The neat PUF has a moisture-absorption range of 1.1–2.5% after 7 days. The neat PUF is the lowest, whereas adding the WHF enhanced moisture absorption. The PUF/WHF with a lower size of WHF absorbs more than the large size. The PU is normally classified based on its low polarity and water resistance [31]. As a result, it has a low moisture absorption rate. Meanwhile, the addition of WHF results in an open cell and improves the polarity of the PUF composites. Through the open cell, moisture may be delivered into the pore of PUF. It is related to the SEM image of the PUF composites, which exhibits more open cells when the WFH is small compared to when the WHF is large.

#### 3.3.4. Flammability

Table 4 shows the flammability of the PUF and PUF/WHF composites with varying WHF sizes. It takes around 125 s to burn a PUF sample to 125 mm. Meanwhile, the addition of WHF with mesh sizes of 20 and 40 resulted in a decrease in burning time. The addition of WHF with a mesh size of 80, on the other hand, can increase flame resistance. It takes around 214 s to reach 125 mm. One of the possibilities is that the addition of WHF caused the cell to open, allowing oxygen to flow to the cell. It can effectively use air to support combustion and make a flame spread quickly [32]. However, when fine fiber is burned, the ashes obscure the fire path and make it more difficult to ignite. As a result, PUF/WHF-80 is the most flame resistant.

#### 3.3.5. Sound Absorption Properties

The sound absorption coefficient (SAC) was used to assess the sound absorption qualities throughout a frequency range of 500 to 6000 Hz. Figure 7 shows the results of evaluating the PUF/WHF composites compared to the commercial PUF, PUF/glass fiber, and polyester foam. Any material having an SAC greater than 0.4 is defined sound-absorbing material [33]. The average SAC of the PUF composites was more than 0.5, with the PUF-WHF-80 obtaining the highest. The efficiency of the sound absorption could be described by the cell foam number. According to Ji et al. [30], the sound-absorption performance of PUF was controlled by the number of pores, and the more the pores, the greater the sound-absorption performance. Furthermore, the inclusion of low-density material could enhance the SAC at a low frequency range. As a result, the PUF with WHF-80, which has the lowest density and the most cell pores, was the best possible sound-absorbing material. Furthermore, the two-peak tendency on the SAC was discovered in the prepared PUF composites. It is because there are varied cell sizes in the PUF composite and the trend of the SAC spectra is comparable to the findings of Chen and Jiang [22].

In comparison, the commercial PUF and PUF/glass fiber had SACs of 0.4, whereas the polyester foam approached 0.85 in the high frequency range. The prepared PUF composites are therefore superior sound-absorbing materials compared to the commercial ones, based on the efficiency of the SAC findings. Furthermore, PUF-WHF-80 was a suitable sound-absorbing material derived from bio-mass with a high potential for sound absorption and is a prospective option for industrial green manufacturing, with the highest SAC value of 0.92.

## 4. Conclusions

The high efficiency of the UPO-based PUF sound-absorbing material, with an NCO index of 100 and 1%wt of WHF, was successfully developed. The size of a fiber has a significant impact on its morphology, mechanical characteristics, flammability and sound absorption. The manufacture of sound-absorbing material with an SAC value of 0.92 was appropriate for the PUF with an 80 mesh size. Furthermore, a biomass-based substance generated from discarded water hyacinth and UPO might be a promising candidate for the sound-absorbing material industry.

## Figures and Tables

**Figure 1 polymers-14-00201-f001:**
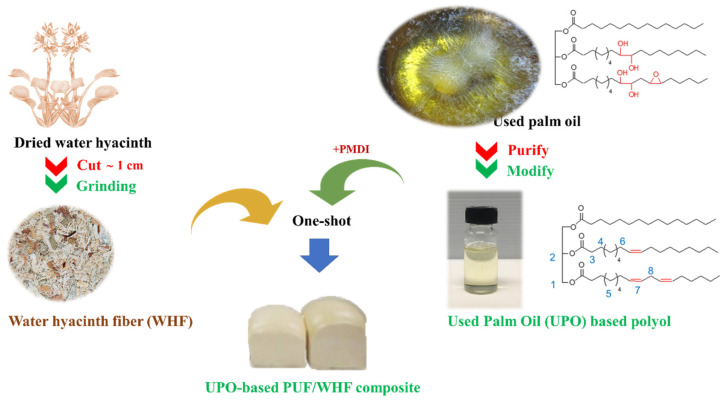
A brief overview of the PUF composite preparation process.

**Figure 2 polymers-14-00201-f002:**
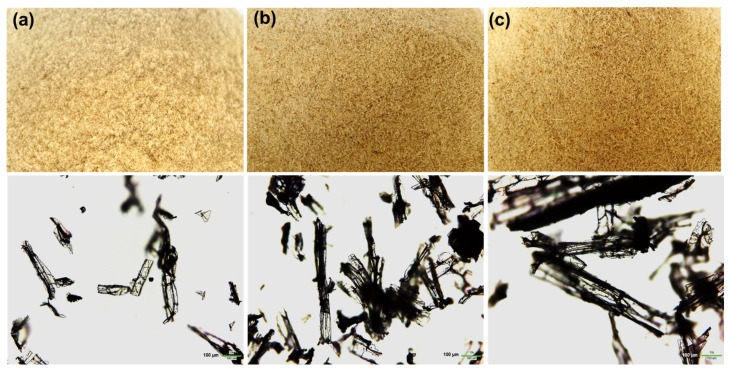
Physical appearances (upper) and optical microscope images (lower) of WHF with different mesh size—(**a**) 80 mesh, (**b**) 40 mesh and (**c**) 20 mesh, respectively.

**Figure 3 polymers-14-00201-f003:**
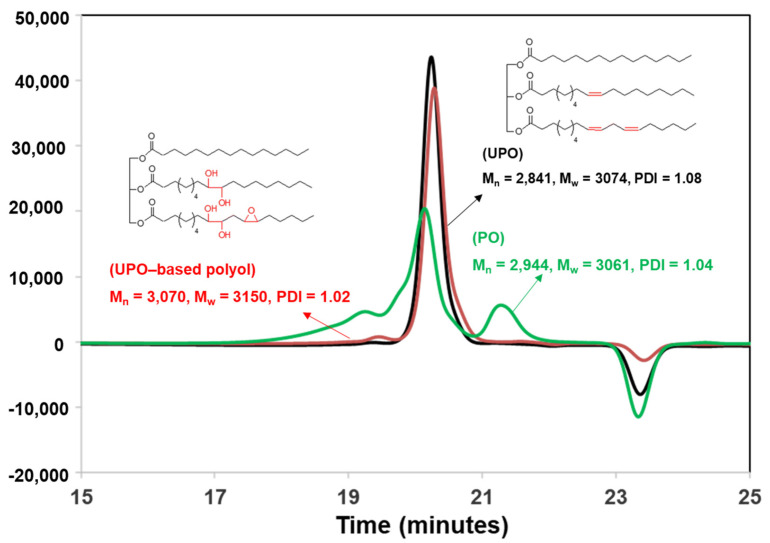
SEC traces of the PO, UPO and UPO−based polyol.

**Figure 4 polymers-14-00201-f004:**
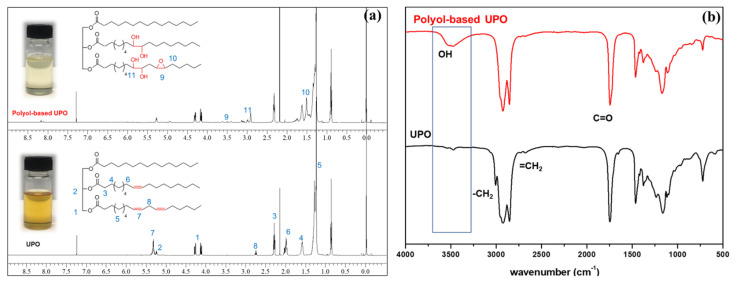
(**a**) ^1^H−NMR spectra and (**b**) FTIR spectra of UPO and polyol−based UPO.

**Figure 5 polymers-14-00201-f005:**
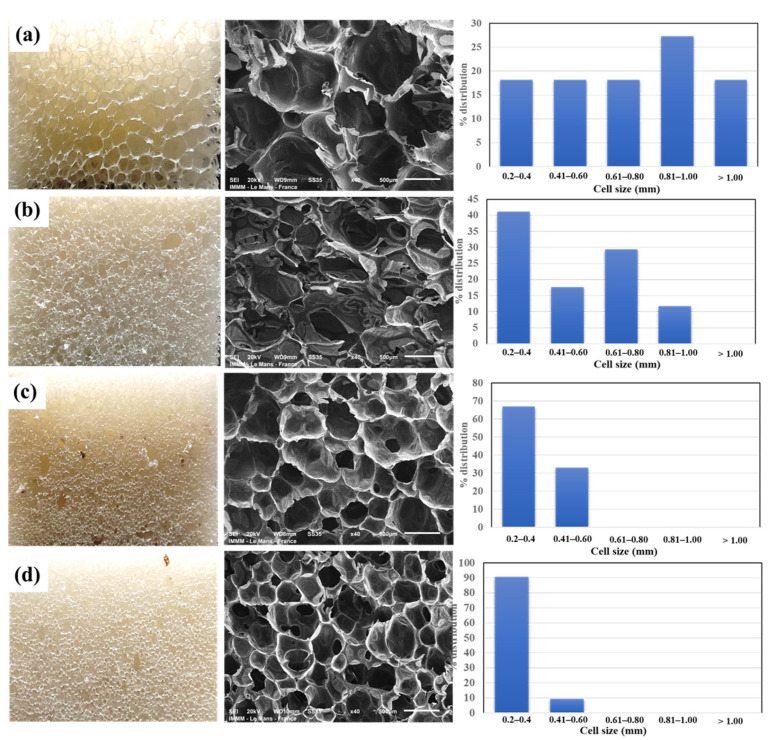
(left) Physical appearances, (middle) SEM images of PUF composite series (at a magnification of ×30) and (right) histograms of their cell diameter distributions of PUF composites—(**a**) PUF, (**b**) PUF-WHF-20, (**c**) PUF-WHF-40, and (**d**) PUF-WHF-80, respectively.

**Figure 6 polymers-14-00201-f006:**
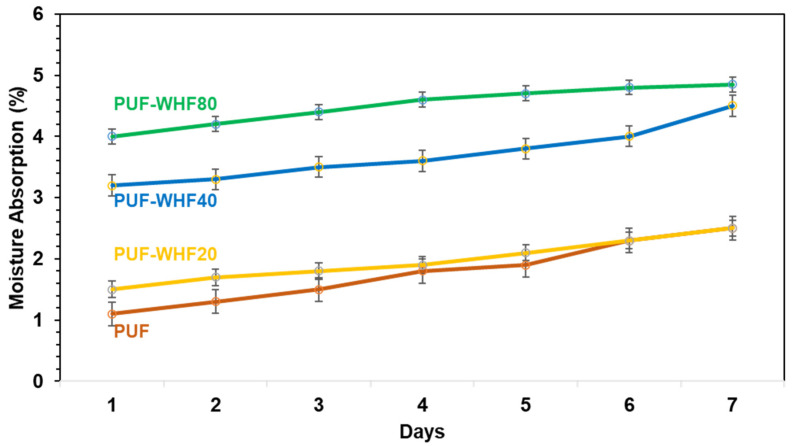
The moisture absorption of PUF and PUF/WHF composite over 7 days.

**Figure 7 polymers-14-00201-f007:**
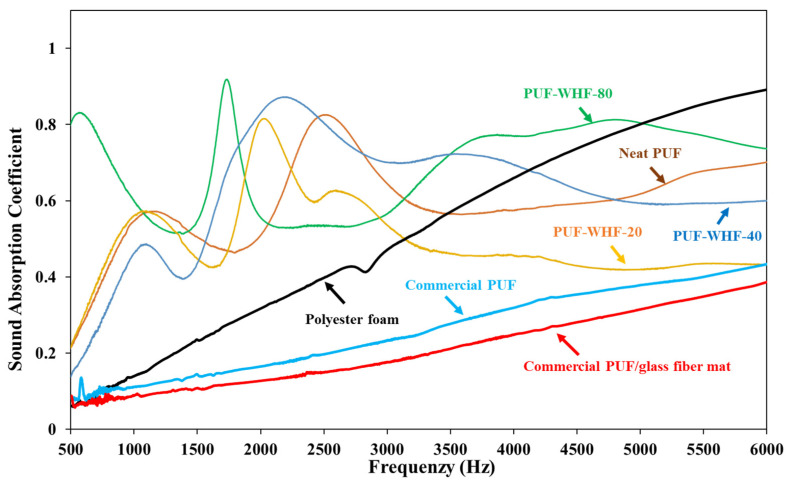
SAC spectra of PUF/WHF composites series compared with commercial sound absorption materials.

**Table 1 polymers-14-00201-t001:** Formulations of UPO-based PUF/WHF composite.

Sample Name	PUF Precursors (mol)			Additives (%wt)		
	RPO	H_2_O	PMDI	WHF-20	WHF-40	WHF-80
PUF	1	1	2	-	-	-
PU-WHF-20	1	1	2	1	-	-
PU-WHF-40	1	1	2	-	1	-
PU-WHF-80	1	1	2	-	-	1

**Table 2 polymers-14-00201-t002:** Properties of UPO and UPO-based polyol.

Sample Name	Iodine Number	OH Value (mg. KOH/g)	Acid Number (mg. KOH/g)	Molecular Weight by SEC		
				*M_n_* (g/mol)	*M_w_* (g/mol)	PDI
UPO	40.1	0	1.41	2841	3074	1.08
UPO-based polyol	0.51	192.19	1.76	3073	3150	1.02

**Table 3 polymers-14-00201-t003:** Properties of RPO-based PUF/WHF Composites.

Sample Name	Cream Time (s)	Rise Time (s)	Track Free Time (s)	Height (cm)	Density (g/mL)	Hardness (Shore OO)	Compressive Strength (kPa)
PUF	12	23	1282	8.4	0.095	29	0.027 ± 0.003
PUF-WHF-20	14	35	1154	6.2	0.066	33	0.047 ± 0.005
PUF-WHF-40	14	32	1205	7.5	0.062	37	0.042 ± 0.007
PUF-WHF-80	13	27	1255	8.0	0.061	45	0.033 ± 0.003

**Table 4 polymers-14-00201-t004:** Flammability of PUF and PUF/WHF composites.

Sample	Burning Time (s)		
	25 mm	60 mm	125 mm
PUF	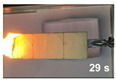	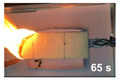	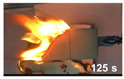
PUF-WHF-20	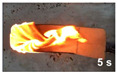	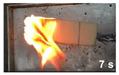	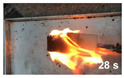
PUF-WHF-40	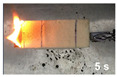	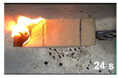	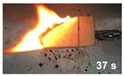
PUF-WHF-80	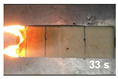	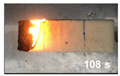	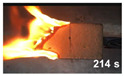

## Data Availability

The study did not report any data.

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
