# Peer review of "Sustainable Rigid Polyurethane Foam from Wasted Palm Oil and Water Hyacinth Fiber Composite—A Green Sound-Absorbing Material"

_polymers, 2022, doi:10.3390/polym14010201_

Round 1

Reviewer 1 Report

Explain better how you got the different samples, explain better how you got the different processing phases.
How many samples have you made?

Figure 6

Describe the characteristics of the impedance tube.
The data of flow resistance and porosity are missing
For each sample, how many measurements have you made?
Do not report low frequency sound absorption values. In the graph you have fluctuating values of the absorption coefficient.
The graph of Figure 6, it is not clear, shows the values from 300 Hz upwards.
Also you have to explain the behavior of the absorption coefficient: PU-WHF-80; PU-WHF-20 and PU-F, have a double bell trend, you have a maximum, then the absorption coefficient decreases, can you explain this effect?
The discussion paragraph is missing.
The number of referecens increases, see also Asbrubali, Arenas, Iannace, Ciaburro, Hyroshnkov.

Author Response

On behalf of the authors, we would like to thank the editor and the reviewers for their informative recommendations and remarks. We revised the manuscript in response to the suggestions. Our responses are included in the table below. In the revised manuscript, these revisions are highlighted in red (using track change).

Reviewer 2 Report

General comments
Good work but badly written manuscript. Major revisions required.

Specific comments
Introduction is pretty weak and needs thorough rewrite. Obviously this is not the first report on synthesis of PU from palm oil. There are many reports on PU foams from palm oil which must be properly cited. Rather than littering introduction with inappropriate self citations authors must focus their efforts on providing a proper detailed literature about synthesis of PU from palm oil and how this work will provide improvements/additions to existing procedures.
Remove unnecessary self citations, especially 6, 7 and 8. This manuscript is on palm oil foam and the self cited papers are on PU adhesives and are no way concerned with this work. Citations are good but stooping to this low is border line unethical practice. 
This reviewer has serious questions about validity of NMR data. Polyols synthesized from used vegetable oil always contain nitrogen and/or sulphur moities but in this work there are none. Why? Explain.
More characterization about "water hyacinth fiber (WHF)" must be given. Authors started with "1 cm length of dried water hyacinth" and ground it down to 40 and 80 mesh size. It should be called as "water hyacinth powder" rather than WHF. No doubt this so-called fibers are absent both in optical and SEM images.
There are many more problems like misleading statements like "generate the PUF more open cell". Except in a and b, SEM images show no or very low percentage of open cells!!! No scale bar in Fig. 4. a,b,c and d. Relabel all images.

Author Response

(The authors gave the same response as above.)

Round 2

Reviewer 2 Report

Authors have made necessary corrections in revised manuscript. But despite reminding authors have not specified scale bar in Fig. 5 a, b, c and d. Paper can be accepted after this minor correction. 

Author Response

On behalf of the authors, we would like to thank the editor and the reviewers for their informative recommendations and remarks. We revised the manuscript in response to the suggestions. Our responses are included in the table below. In the revised manuscript, these revisions are highlighted in red (using track change) as shown in attached files.

With my best regards

Sincerely yours,

Nathapong SUKHAWIPAT
